# A Clinical Case of Scrub Typhus in the United States Forces Korea Patient with Eschar and Genetic Identification of *Orientia tsutsugamushi* Using Multiplex PCR-Based Next-Generation Sequencing

**DOI:** 10.3390/pathogens10040424

**Published:** 2021-04-02

**Authors:** Seungchan Cho, Jon C. Allison, Kkothanahreum Park, Jin Sun No, Seung-Ho Lee, Kyungmin Park, Jongwoo Kim, Terry A. Klein, Heung-Chul Kim, Won-Keun Kim, Jin-Won Song

**Affiliations:** 1Department of Microbiology, College of Medicine, Korea University, Seoul 02841, Korea; schanchan@korea.ac.kr (S.C.); pkhar@korea.ac.kr (K.P.); njs2564@gmail.com (J.S.N.); leeds1104@korea.ac.kr (S.-H.L.); kmpark0131@korea.ac.kr (K.P.); hotdog442@korea.ac.kr (J.K.); 2Force Health Protection and Preventive Medicine, MEDDAC-Korea, 65th Medical Brigade, Unit 15281, APO AP 96251-5281, USA; jon.c.allison.mil@mail.mil (J.C.A.); terry.a.klein2.civ@mail.mil (T.A.K.); hungchol.kim2.ln@mail.mil (H.-C.K.); 3Department of Microbiology, College of Medicine, Hallym University, Chuncheon 24252, Korea; wkkim1061@hallym.ac.kr; 4Institute of Medical Science, College of Medicine, Hallym University, Chuncheon 24252, Korea

**Keywords:** *Orientia tsutsugamushi*, scrub typhus, genetic identification, target enrichment next-generation sequencing

## Abstract

An epidemiological investigation was conducted for a scrub typhus case reported in a U.S. Forces Korea military patient in the Republic of Korea in November 2018. The patient had a fever, maculopapular rash, and an eschar. The full-length sequence of *Orientia tsutsugamushi* 56-kDa type-specific antigen (TSA) gene was obtained from eschar tissue by multiplex PCR-based Next Generation Sequencing for genetic identification. Based on the 56-kDa TSA gene, the *O. tsutsugamushi* aligned most closely with the Boryong strain.

## 1. Introduction

Scrub typhus is an acute febrile bacterial disease caused by the arthropod-borne gram-negative bacillus, *Orientia tsutsugamushi*, commonly identified in rural areas. Scrub typhus is endemic to the Asia-Pacific region, extending from Afghanistan to Korea, China, the islands of the Southwestern Pacific, and Northern Australia, and might also be present in Africa and South America [1,2,3]. In recent years, there have been approximately 8000 clinical cases of scrub typhus annually in Korea, with a fatality rate of 1.32% with good medical care [4]. Typically, clinical manifestation and signs develop within one to two weeks after infection and are characterized by fever, headache, muscle pain, cough, gastrointestinal upset, maculopapular rash, and eschar. Recently, the genetic analysis of *O. tsutsugamushi* has been used to establish and differentiate genotypes [5]. More than 20 antigenic variants of *O. tsutsugamushi* have been reported worldwide, including Boryong, Kuroki, Gilliam, Kato, Karp, and Kawasaki strains.

The immune-dominant 56-kDa type-specific antigen (TSA) is located on the surface of the bacteria membrane [6]. The 56-kDa TSA is not expressed in other bacteria, including other members of the *Rickettsiaceae*. Other genes may be useful for studies of differentiation within *Orientia* strains, but to date, no other locus seems to have the same potential of usefulness for analysis of strain variation [5]. Compared to the genes for 16S rRNA or groEL, the high level of variability of 56-kDa TSA appears to be a representative marker gene for differentiating *O. tsutsugamushi* strains [7]. Target-enrichment nextgeneration sequencing (NGS) methods exhibited the effective recovery of viral genome sequences from rodent tissue containing low copies of viral RNA [8,9]. To investigate an extremely low amount of genome sequences of *O. tsutsugamushi* from the patient’s tissue (eschar) with scrub typhus, we performed multiplex PCR-based NGS for the enrichment of the 56-kDa TSA gene. The obtained full-length sequences allowed the genetic identification of *O. tsutsugamushi* genotypes. Phylogenetic analysis of the full-length 56-kDa gene demonstrated that the United States Forces Korea (USFK) patient was infected with *O. tsutsugamushi* that aligned with the Boryong strain. The clinical aspect of the scrub typhus USFK patient and genetic characterization of *O. tsutsugamushi* raises awareness for the surveillance of an endemic zoonotic disease, scrub typhus, in the ROK.

## 2. Results

From 3–4 November 2018, a 46-year-old male and his family visited Jirisan National Park located at the southern tip of the Korean Peninsula. He visited a gorge, temple site, and garden area where he walked along trails in a forested area with cut/uncut grasses/herbaceous vegetation. On 12 November 2018, the patient noted a nickel-sized black scab on his lower left calf of his leg without other symptoms. On 14 November, the patient felt very tired after running and a day later experienced generalized myalgia with mild intermittent headaches and sweats (Table 1). He took Ibuprofen for pain relief (PO 200–400 mg PRN).

On 16 November 2018, the patient reported to the Jenkins Army Health Clinic, Camp Humphreys, with a chief complaint of a “bug bite” (eschar) about the size of a nickel with a localized inflammatory reaction on his lower left calf (Figure 1A). He reported myalgia with mild intermittent headaches, chills, and sweats. He was prescribed acetaminophen, PO 325 mg, 1 to 2 tablets every 4 h as needed for pain or fever. On 17–18 November, he continued to have intermittent headaches, chills, and night sweats.

On 19 November 2018, the patient noted a body rash and reported to the Jenkins Army Health Clinic with worsening symptoms of intermittent dizziness, fever, chills, night sweats, fatigue, and loss of appetite. He noted a “non-itchy” maculopapular rash that developed over the entire body 24 h prior to the visit (Figure 1B). He was diagnosed with an insect bite (nonvenomous) that was red and raised with a dark center on his left lower leg and pruritus and a maculopapular rash of unclear etiology. The patient was prescribed sulfamethoxazole/trimethoprim (800 mg/160 mg), 1 tablet every 12 h, to treat a presumptive bacterial infection. On the following day (20 November), he noted that the rash had spread from the body to the legs. On 21 November, the patient indicated that he felt better, but the body rash increased.

On 21 November, the patient was referred to the Saint Mary’s Hospital, Pyeongtaek where he was admitted with fever, chills, myalgia, maculopapular rash (head, trunk), sweats, and eschar, that had been presumptively diagnosed as an insect/spider bite. Based on the symptoms, the patient was presumptively diagnosed with scrub typhus and prescribed doxycycline, 100 mg 3 times daily after a meal for treatment of *O. tsutsugamushi*, acetaminophen 650 mg for pain, acetylcysteine 200 mg to reduce liver damage resulting from extensive use of acetaminophen, ursodeoxycholic acid 250 mg to reduce liver damage, Godex 150 mg capsule (carnitine orotate 150 mg, 17 amino acids, 12.5 mg; pyridoxine HCl, 25 mg; adenine HCl, 2.5 mg, cyanocobalamin 125 μg, and riboflavin, 500 μg) to reduce liver enzyme level, and rebamipide 100 mg to provide gastrointestinal protection.

On 26 November, the patient was interviewed by Force Health Protection and Preventive Medicine, 65th Medical Brigade, and a biopsy of the eschar was obtained with the patient’s verbal approval. The eschar tissue was placed in a −80 °C freezer and transported to Korea University the following day. The patient provided photos of the rash and eschar, as well as area information of the military base where he worked and Jirisan National Park that he visited (Figure 1C). On 28 November, the fever had subsided, and the patient was discharged from the hospital.

We extracted DNA from the eschar and performed nested PCR (the expected size of an amplicon, 632 bps) using specific primers for the 56-kDa TSA (1599 bps) of *O. tsutsugamushi*. The phylogeny of the 56-kDa TSA nucleotide sequence formed a close genetic lineage with the Boryong genotype, showing 99.8% (631/632). The amino acid sequence of 56-kDa TSA also showed a high homology of 99.5% with the Boryong genotype.

To obtain the full-sequence of 56-kDa TSA gene from the patient’s eschar tissue containing an extremely low amount of the *O. tsutsugamushi* genome, we designed multiplex PCR primers with 150 or 300 bps interval and amplified the targeted gene as shown (Table 2) and performed the NGS using MiSeq Benchtop. The complete genomic sequence of 56-kDa TSA from the patient was inferred to a phylogenetic group belonging to the Boryong strain (Figure 2). The genomic sequence was deposited in the GenBank (accession number: MT989475).

## 3. Discussion

Soldiers from ROK and the U.S. are exposed to rodents and their associated ectoparasites that may harbor zoonotic pathogens of military importance. Exposure to trombiculid mites that may be infected with *O. tsutsugamushi* increases potential disease risks during military operations at installations and training sites located throughout the Korean Peninsula. Investigations to identify infection rates of scrub typhus in USFK personnel have been carried out routinely through the comprehensive rodent field surveillance program [10]. While few scrub typhus cases (1 each year for 1995 and 2003, 2 for 2012, and 1 for 2013) were identified among US soldiers, it remains a serious health threat as it can quickly incapacitate large numbers of people and undermine military operations [11]. The serological survey of rodents, collected in the ROK, including the US operated military training sites located near the Demilitarized zone, demonstrated that the prevalence of *O. tsutsugamushi* showed a sero-positivity of 45.6%, 23.1%, and 25.0% in *Apodemus agrarius*, *Mus musculus*, and *Rattus norvegicus*, respectively [12]. Therefore, continuous rodent trapping and surveillance are needed to identify disease risks in preventing the occurrence of patients with scrub typhus.

Using a multiplex PCR-based NGS strategy, we obtained the complete genomic sequence of 56-kDa TSA for *O. tsutsugamushi* for an eschar of the USFK patient. To investigate the emerging genotypes, a phylogenetic tree was constructed using the 56-kDa TSA full sequences. The result demonstrated that the USFK patient was infected with the Boryong genotype of *O. tsutsugamushi*, which is the most prevalent strain in the southwest part of ROK [6]. The clinical genotype of *O. tsutsugamushi* accounted for the Boryong strain of 79% and the Karp strain of 15%, respectively, in ROK. Thus, we confirmed the USFK patient was most likely infected with the Boryong strain of *O. tsutsugamushi*.

In conclusion, the diagnosis and treatment of the USFK patient with scrub typhus were achieved by clarifying an eschar, clinical manifestations, and conducting an epidemiological survey. The multiplex-PCR-based NGS allowed for the delineation of the genotype of the clinical *O. tsutsugamushi* strain by acquiring the complete genomic sequence of 56-kDa TSA directly obtained from the clinical specimen. The clinical aspect of the USFK patient with scrub typhus and genetic identification of *O. tsutsugamushi* provides significant insights into clinical manifestations, epidemiological interviews, and high-throughput sequencing for the preparedness of the bacterial disease. This report raises awareness among physicians for the zoonotic disease, scrub typhus, especially in high endemic regions of the ROK.

## 4. Materials and Methods

### 4.1. DNA Isolation from Eschar and Nested PCR

Bacterial DNA was obtained from eschar by standard procedures using a High Pure PCR Template Preparation Kit (Roche, Basel, Switzerland). Nested PCR was performed in a 25 μL reaction mixture containing 0.1 mM dNTP Mix, 0.625 units TaKaRa Ex Taq polymerase (Takara, Shiga, Japan), 0.4 μM of each primer, and 1.5 μL template. Oligonucleotide primer sequences were Ri-A (outer): 5′-TTTCGAACGTGTCTTTAAGC-3′, Ri-B (outer): 5′-ACAGATGCACTATTAGGCAA-3′, Ri-E (inner): 5′-GTTGGAGGAATGATTACTGG-3′, and Ri-F (inner): 5′-AGCGCTAGGTTTATTAGCAT-3′ for the 56-kDa TSA gene [13]. The PCR conditions were: initial denaturation at 95 °C for 5 min, followed by 30 cycles of denaturation at 94 °C for 40 s, annealing at 52 °C for 40 s, elongation at 72 °C for 90 s. Additionally, final elongation was done at 72 °C for 7 min. PCR products were extracted using a PCR Purification Kit (Cosmo Genetech, Seoul, Korea), and DNA sequencing was performed in an Automatic Sequencer, ABI 3730XL DNA Analyzer (Applied Biosystems, Foster City, CA, USA).

### 4.2. Multiplex PCR-Based Next-Generation Sequencing (NGS)

Multiplex PCR primers were designed with 150 or 300 bps intervals using BioEdit Sequence Alignment Editor (Version 7.1.11). Each primer for the multiplex PCR of *O. tsutsugamushi* 56-kDa TSA gene was prepared and mixed (Table 2). The first PCR was performed in 25 μL reaction mixture containing 12.5 μL 2× Uh pre-mix, 1.0 μL of genomic DNA template, 1.0 μL of each primer mixture, and 10.5 μL distilled water (DW). Initial denaturation was performed by a cycle of 95 °C for 15 min, then 40 cycles of 95 °C for 20 s, 50 °C for 40 s, 72 °C for 1 min, and a cycle of final elongation at 72 °C for 3 min. The second PCR was performed in 25 μL reaction mixture containing 12.5 μL 2× Uh pre-mix, 1.0 μL of the first PCR product, 1.0 μL of each primer mixture, and 10.5 μL DW. Initial denaturation was performed at 95 °C for 15 min, then 25 cycles of 95 °C for 20 s, 50 °C for 40 s, 72 °C for 1 min, and final elongation at 72 °C for 3 min. DNA libraries were prepared using a TruSeq Nano DNA LT sample preparation kit (Illumina, San Diego, CA, USA) according to the manufacturer’s instructions. The samples were mechanically sheared using an M220 focused ultrasonicator (Covaris, Woburn, MA, USA). The amplicon was size-selected, A-tailed, ligated with indexes and adaptors, and enriched. The libraries were sequenced using a MiSeq benchtop sequencer (Illumina) with 2 × 150 bp with a MiSeq reagent V2 (Illumina).

### 4.3. Phylogenetic Analysis

The genome sequence of the *O. tsutsugamushi* 56-kDa TSA gene was aligned using the Clustal W method with the Lasergene program, version 5 (DNASTAR). Phylogenetic analysis was conducted using the Bayesian inference method. Topologies were evaluated by bootstrap analysis of 1000 iterations.

## Figures and Tables

**Figure 1 pathogens-10-00424-f001:**
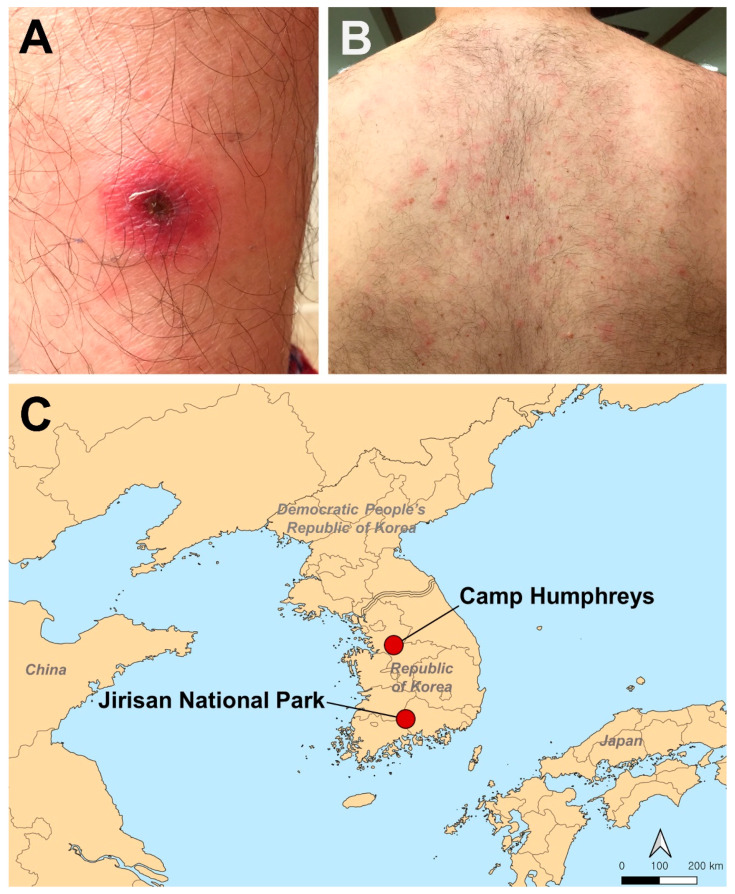
An eschar on the lower left lower calf and maculopapular rash on the back of the USFK patient with *Orientia tsutsugamushi* infection. (**A**) Eschar location; (**B**) Back with maculopapular rash; (**C**) The location of Camp Humphreys and Jirisan National Park, Republic of Korea.

**Figure 2 pathogens-10-00424-f002:**
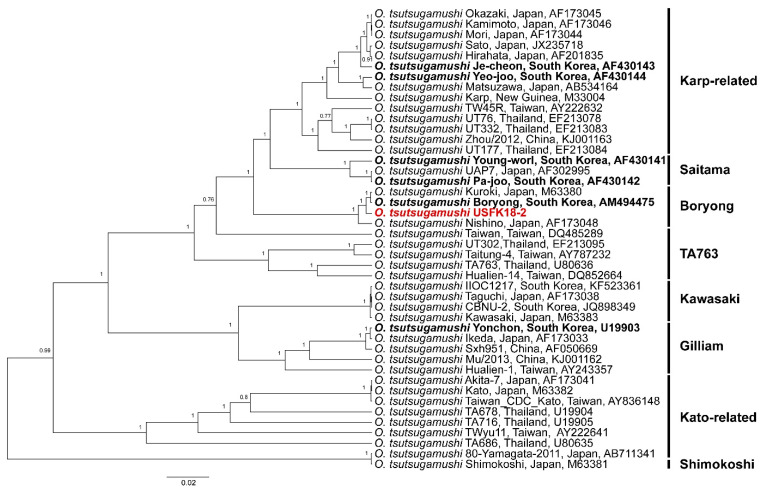
Phylogenetic analysis of the *O. tsutsugamushi* 56-kDa TSA gene identified from the USFK patient eschars. Phylogenetic relationships of the whole-genome sequences of *O. tsutsugamushi* 56-kDa TSA gene were inferred using Bayesian inference in BEAST (v1.10.4), using default priors and assuming homochromous tips. Upon running the Markov chain Monte Carlo analyses until adequate sample sizes (ESS > 200) were obtained, TreeAnnotator (v2.5.4) was used to summarize a maximum clade credibility tree from the posterior tree distribution, using a 10% burn-in. The Korean serotypes, Je-cheon (AF430143), Yeo-joo (AF430144), Pa-joo (AF430142), Young-worl (AF430141), Boryong (NC009488), and Yonchon (U19903), are indicated in bold. Topologies were evaluated by bootstrap analysis of 1000 iterations. Numbers along branches are bootstrap values. The scale bar represents nucleotide substitutions per site.

**Table 1 pathogens-10-00424-t001:** Clinical signs and symptoms of the United States Forces Korea (USFK) patient infected by *O. tsutsugamushi.*

Date	Onset/Clinical Visit	Antibiotics	Eschar	Rash	Symptoms							
					Chills	Fever	Sweats	Headache	Nausea/Vomiting	Diarrhea	Body Aches	Malaise
12 November 2018	Eschar	No	Yes	No	No	N/D *	No	No	No	No	No	No
14 November 2018	Onset	No	Yes	No	No	N/D	Yes	Yes	No	No	No	Yes
15 November 2018	Continued	No	Yes	No	Yes	N/D	Yes	Yes	No	No	Yes	Yes
16 November 2018	Cp Humphreys Clinic	No	Yes	No	No	98.1 °F	Yes	Yes	No	No	Yes	Yes
19 November 2018	Cp Humphreys Clinic	No	Yes	Yes	Yes	98.3 °F	Yes	Yes	loss of appetite	No	Yes	Yes
21 November 2018	St Mary’s Hospital	Doxycycline **	Yes	Yes	Yes	97.9 °F	Yes	Yes	No	No	Yes	Yes
25 November 2018	St Mary’s Hospital	Doxycycline	Yes	Yes	No	N/D	No	No	No	No	Yes	Yes
26 November 2018	St Mary’s Hospital	Doxycycline	Yes	Yes	No	N/D	No	No	No	No	Yes	Yes

* N/D, not determined. ** Doxycycline (100 mg) was prescribed twice daily.

**Table 2 pathogens-10-00424-t002:** Primer sequences used for multiplex PCR-based next-generation sequencing.

Name	Position	Sequence (5′→3′)	Reference	Name	Position	Sequence (5′→3′)	Reference
56kDa-01F	1–20	CTAGAAGTTATAGCGCACAC	Boryong (AM494475)	56kDa-01R	281–300	GATCTGAGTATGGTTGTTGG	Boryong (AM494475)
56kDa-02F	251–270	GTAACTAGGTCAGCATAGAG		56kDa-02R	531–550	GCAAGCTCAAGCTACAGCGC	
56kDa-03F	501–520	GCCTAACAGCTGCTGCTGCT		56kDa-03R	781–800	AAATTATTCAGATATATAGT	
56kDa-04F	751–770	ACCAGCTATATCAGCAAATG		56kDa-04R	1031–1050	AAGAATTGTGCTGGTATTGA	
56kDa-05F	1007–1020	CCAGGATTATTAGGATCCTT		56kDa-05R	1281–1300	CGCTGAGGTTGAAGTAGGTA	
56kDa-06F	1251–1270	TTATCTCACCTTTAGAATCT		56kDa-06R	1531–1550	CGTTTTCAGCTAGTGCGATA	
56kDa-07F	1501–1520	ACACTCTAATCCTACTTCAT		56kDa-07R	1577–1599	ATGAAAAAAATTATGTTAATTGC	
56kDa-08F	1–20	CTAGAAGTTATAGCGTACAG	Young-worl (AF430141)	56kDa-08R	131–150	GCATCAGGGGCGCTTGGTGT	Young-worl (AF430141)
56kDa-09F	101–120	ACATACACACCCTCAGCAGC		56kDa-09R	231–250	AACTGAATCATTCTCAATAT	
56kDa-10F	201–220	TATGAGCTAACCCTGCACCA		56kDa-10R	331–350	GTAGCTGCAAGAAGGRCGAT	
56kDa-11F	301–320	AAACTCTGTTTCTTTGCTTW		56kDa-11R	431–450	TTGAAGCGTCATGCAGGATT	
56kDa-12F	401–420	TGAGCAGCTAATTTMTCCAT		56kDa-12R	631–650	YTTTTGAKGGGTATATTRGT	
56kDa-13F	601–620	GACAAAATTCAACTGTATCT		56kDa-13R	731–750	CCTAATRSTGCTTTGCCTAA	
56kDa-14F	701–720	ATCTGTTCGACAGATGCACT		56kDa-14R	831–850	AATAAACCTAGCGATCCTCC	
56kDa-15F	801–820	TATCACTTAATACTTTGACA		56kDa-15R	931–950	ATCCTGCTGGAAATCCACCG	
56kDa-16F	901–920	CGCAAAATCTGCAGGCTGAT		56kDa-16R	1031–1050	AATTGTGCTGGTATTGACTA	
56kDa-17F	1001–1020	CCATTAGGATTATTAGGATC		56kDa-17R	1131–1150	TTGCGGTTGATATTCCTAAC	
56kDa-18F	1101–1120	GATTTGCAGCTTGCGCTTGC		56kDa-18R	1231–1250	ATTCTGGAGGTGGGACAGAT	
56kDa-19F	1201–1220	AAGTTTAAACCGCTTACGTA		56kDa-19R	1331–1350	CCATGGCTTAGAGCAGAGCT	
56kDa-20F	1301–1320	GCGCTTATATTTCTAAGGTA		56kDa-20R	1431–1450	TGCTCGCTTGGATCAAGCTG	
56kDa-21F	1401–1420	TGTACCACCAAATGGCATCG		56kDa-21R	1531–1550	GATGAAGGAGGATTAGAGTG	
56kDa-22F	1501–1520	CCGACAACTCCAACTTTAGC					

## Data Availability

All data are included in the manuscript. The sequence generated during the current study is available at GenBank (accession number: MT989475).

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
