# Peer review of "A Clinical Case of Scrub Typhus in the United States Forces Korea Patient with Eschar and Genetic Identification of *Orientia tsutsugamushi* Using Multiplex PCR-Based Next-Generation Sequencing"

_pathogens, 2021, doi:10.3390/pathogens10040424_

Round 1

Reviewer 1 Report

This is a short and straightforward report of a case of scrub typhus in a patient in Korea. Novelty comes from the PCR amplification and sequencing directly from the eschar. This makes the study interesting and relevant to others in the field. 

I have a number of minor concerns:

  1. Line 30 - should state that these numbers are from Korea. Reads as though they are global numbers.
  2. Table 1. Should add treatments (doxycycline) and are there later time points - when did fever subside? Were there long term symptoms? 
  3. Lines 98-102, authors should state the length of the amplicon as well as the length of the full gene. This helps readers understand why they went on to do more targeted sequencing.
  4. Lines 103-109 and Fig 2. Authors could describe more clearly the positions of the primers on the gene. i.e. are the amplicons overlapping? Do they cover the entire gene? A figure showing the positions of primer pairs would be useful.
  5. Line 163. Please double check the sequence of Ri-B (outer). By my analysis this binds inside gene OTBS_1802 which seems unlikely.
  6. I am not sure the title is correct. Target enrichment NGS refers to cases where targets are physically enriched on beads or by other means. Here the target was amplified by PCR which is not the same thing. This seems misleading. 

Author Response

  1. overlapping? Do they cover the entire gene? A figure showing the positions of primer pairs would be useful.

R: We added more information about the position of the primers on the gene and references in Table 2. This primer sets cover the entire 56-kDa TSA gene. We designed multiplex PCR primers with 150 or 300 bps interval and amplified the targeted gene. Also, amplicons are overlapped.

  1. Line 163. Please double check the sequence of Ri-B (outer). By my analysis this binds inside gene OTBS_1802 which seems unlikely.

R: We cited a reference about the primer information for Nested PCR. We checked the sequences of primers between our manuscript and the reference.

  1. I am not sure the title is correct. Target enrichment NGS refers to cases where targets are physically enriched on beads or by other means. Here the target was amplified by PCR which is not the same thing. This seems misleading.

R: As your comment, we changed the title for specifying the meaning as shown below.

Title: A Clinical Case of Scrub typhus in The United States Forces Korea Patient with Eschar and Genetic Identification of Orientia tsutsugamushi using Multiplex PCR-based Next-generation Sequencing

Reviewer 2 Report

This study reports a clinical case of scrub typhus in South Korea, along with the complete genomic sequence. Given that there are thousands of scrub typhus cases in this country, it’s unclear how novel this study is. The authors should state explicitly how this study differs from the previous studies so that their importance can be justified. Otherwise, it should instead be published in a local journal.

Author Response

Revision

First, we want to thank the reviewers for the helpful comments on our original submission. We have modified the figure and data presentations, and this has certainly resulted in an improved manuscript. We have marked with highlighted and burgundy color font those sections of the manuscript that directly answer the reviewers concerns. Below is a detailed point by point response to the reviewer’s comments.

Reviewer 2’s Comments:

This study reports a clinical case of scrub typhus in South Korea, along with the complete genomic sequence. Given that there are thousands of scrub typhus cases in this country, it’s unclear how novel this study is. The authors should state explicitly how this study differs from the previous studies so that their importance can be justified. Otherwise, it should instead be published in a local journal.

R: This report is important to further understanding of the potential risks that US military personnel can encounter in the Republic of Korea. This study confirmed other studies suggesting that Boryong strain is the most prevalent strain. The study may guide public health monitoring in this area and could help in a diagnosis of scrub typhus in patients that present with these mild symptoms. This report arises awareness among physicians for a zoonotic disease, scrub typhus, especially in high endemic regions of the ROK.

For diagnosis and detection of various viruses, several NGS methods have been developed, including enrichment of viral nucleic acids by using target-specific oligonucleotide probes, removing host genome sequences, purifying virus-like particles, and performing small-RNA deep sequencing (Quick, J. et al. 2017.; Andersen, K.G. et al. 2015.; Reyes, G.R. et al. 1991.; Djikeng, A. et al. 2008.). This study showed that the multiplex PCR-based NGS method is a potential case for application of targeted NGS strategies not only in viruses but also in bacterial pathogen.

Reviewer 3 Report

This is a brief report describing a case of suspected (and subsequently) confirmed scrub typhus in a US military person on duty in South Korea.  This is a very good description of the clinical case, the progression of disease and ultimately the diagnosis of the infection by O. tsutsugamushi strain Boryong in the infected individual. 

Strengths:  The major strength is the use of next-gen sequencing capabilities to rapidly detect and identify O. tsutsugamushi from DNA isolated from the patient's eschar and to use this information to genotype the strain (Boryong) likely causing the infection.  From an epidemiological standpoint,this is important so as to further understand the potential risks that US military personnel can encounter in this part of the world (South Korea).  The study also tends to confirm other studies suggesting that strain Boryong is the most prevalent strain in this part of the world.  The study may also guide public health monitoring in this area and could help in a more timely diagnosis of scrub typhus in patients that present with these mild flu-like symptoms.

Weaknesses:  It is unclear why only a single gene (tsa56) was utilized for genotyping and no rationale is provided for this in the study.  This study and the conclusions would be bolstered by the sequencing of several genes and bioinformatic analyses (as provided for TSA56) to ensure that the proposed genotype of the etiologic agent is correct.

Was it possible to isolate and culture O. tsutsugamushi from the eschar of the infected patient?  The authors make a comment in the manuscript that the genomic DNA isolated from the eschar attributed to O. tsutsugamushi "was low". Therefore, it may not have been feasible (access or lack thereof to BSL3 laboratories is not mentioned).  Again, in the absence of culture (the gold standard), it may be more impactful to the identification of the strain/genotype by sequencing more than one gene.  This may become more important in the future should more virulent/pathogenic strains begin to be identified in regions where US military personnel are deployed.

Author Response

Revision

First, we want to thank the reviewers for the helpful comments on our original submission. We have modified the figure and data presentations, and this has certainly resulted in an improved manuscript. We have marked with highlighted and burgundy color font those sections of the manuscript that directly answer the reviewers concerns. Below is a detailed point by point response to the reviewer’s comments.

Reviewer 3’s Comments:

This is a brief report describing a case of suspected (and subsequently) confirmed scrub typhus in a US military person on duty in South Korea.  This is a very good description of the clinical case, the progression of disease and ultimately the diagnosis of the infection by O. tsutsugamushi strain Boryong in the infected individual.

Strengths:  The major strength is the use of next-gen sequencing capabilities to rapidly detect and identify O. tsutsugamushi from DNA isolated from the patient's eschar and to use this information to genotype the strain (Boryong) likely causing the infection. From an epidemiological standpoint, this is important so as to further understand the potential risks that US military personnel can encounter in this part of the world (South Korea). The study also tends to confirm other studies suggesting that strain Boryong is the most prevalent strain in this part of the world.  The study may also guide public health monitoring in this area and could help in a more timely diagnosis of scrub typhus in patients that present with these mild flu-like symptoms.

Weaknesses: It is unclear why only a single gene (tsa56) was utilized for genotyping and no rationale is provided for this in the study. This study and the conclusions would be bolstered by the sequencing of several genes and bioinformatic analyses (as provided for TSA56) to ensure that the proposed genotype of the etiologic agent is correct.

R: For emphasizing the reason which tas56 was utilized for genotyping of O. tsutsugamushi, we added sentence as shown below in Introduction.

Line 38 – 41: The 56-kDa TSA is not expressed in other bacteria, including other members of the Rickettsiaceae. Other genes may be useful for studies of differentiation within Orientia strains, but to date, no other locus seems to have the same potential of usefulness for analysis of strain variation (5).

We first designed the multiplex PCR primers for several marker genes (47-kDa, 56-kDa, groEL, ScaA, ScaB, ScaD, and ScaE) of O. tsutsugamushi in rodents. Except of tsa56 gene, the genome coverage rate was low. The complete sequences of each genes were not obtained. We agree your comment which is impactful to the identification of genotypes by sequencing using gene candidates. The advance of this NGS methods for genetic identification of O. tsutsugamushi remains to be investigated.

Was it possible to isolate and culture O. tsutsugamushi from the eschar of the infected patient?  The authors make a comment in the manuscript that the genomic DNA isolated from the eschar attributed to O. tsutsugamushi "was low". Therefore, it may not have been feasible (access or lack thereof to BSL3 laboratories is not mentioned).  Again, in the absence of culture (the gold standard), it may be more impactful to the identification of the strain/genotype by sequencing more than one gene. This may become more important in the future should more virulent/pathogenic strains begin to be identified in regions where US military personnel are deployed.

R: In previous study, Rickettsia akari was isolated from eschars of patients with rickettsialpox (Paddock, C. D. et al. 2006). We did not attempt to isolate O. tsutsugamushi from an eschar tissue of the patient due to the permission issue of BSL3 laboratories.

Round 2

Reviewer 2 Report

The research will be of more importance if the infection occurred "inside" the military campus. Otherwise, the chance for people in Korea for encountering with scrub typhus will be similar, whether they are military personnel or not. For a region with high scrub typhus human cases (southern Korea), it is of little surprise for people to infect with scrub typhus. 

Author Response

Reviewer 2’s Comments:

The research will be of more importance if the infection occurred "inside" the military campus. Otherwise, the chance for people in Korea for encountering with scrub typhus will be similar, whether they are military personnel or not. For a region with high scrub typhus human cases (southern Korea), it is of little surprise for people to infect with scrub typhus.

R: As the reviewer’s comment, it is important to identify the sites of suspected infection. In this report, we couldn’t identify the infection site of this USFK patient using genomic epidemiology of 56-kDa TSA in O. tsutsugamushi. Nevertheless, the clinical aspect of the USFK patient with scrub typhus and genetic identification of O. tsutsugamushi provides significant insights into clinical manifestations, epidemiological interview, and high-throughput sequencing for the preparedness of the bacterial disease. There are 30,000 of USFK stationed in Republic of Korea. Investigations to identify infection rates of scrub typhus in USFK personnel have been carried out continuously through the comprehensive rodent field surveillance program. This USFK scrub typhus case arises awareness to physicians for the surveillance of an endemic zoonotic disease, scrub typhus, in the ROK.